# Micro Scalable Graphene Oxide Productions Using Controlled Parameters in Bench Reactor

**DOI:** 10.3390/nano11081975

**Published:** 2021-07-31

**Authors:** Carolina S. Andrade, Anna Paula S. Godoy, Marcos Antonio Gimenes Benega, Ricardo J. E. Andrade, Rafael Cardoso Andrade, Wellington Marcos Silva, Josué Marciano de Oliveira Cremonezzi, Waldemar Augusto de Almeida Macedo, Pedro Lana Gastelois, Helio Ribeiro, Jaime Taha-Tijerina

**Affiliations:** 1MackGraphe–Graphene and Nanomaterials Research Center, Mackenzie Presbyterian University, Rua da Consolação 896, São Paulo 01302-907, Brazil; carolsantosandrade@gmail.com (C.S.A.); annapsgodoy@hotmail.com (A.P.S.G.); marcosagbenega@gmail.com (M.A.G.B.); ricardo.andrade@mackezie.br (R.J.E.A.); josuecremonezzi@hotmail.com (J.M.d.O.C.); 2Escola de Engenharia, Mackenzie Presbyterian University, Rua da Consolação 896, São Paulo 01302-907, Brazil; rafael@4info.com.br (R.C.A.); helio.ribeiro1@mackenzie.br (H.R.); 3Departamento de Química, Universidade Federal de Minas Gerais—UFMG, Avenida Presidente Antônio Carlos, 6.627, Belo Horizonte 31270-901, Brazil; wellingtonmarcos@yahoo.com.br; 4Centro de Desenvolvimento da Tecnologia Nuclear—CDTN, Avenida Presidente Antônio Carlos, 6.627, Belo Horizonte 31270-901, Brazil; wmacedo@cdtn.br (W.A.d.A.M.); pedrogastelois@gmail.com (P.L.G.); 5Departamento de Ingeniería, Universidad de Monterrey, Av. Ignacio Morones Prieto 4500 Pte., San Pedro Garza García 66238, Mexico; 6Department of Manufacturing and Industrial Engineering, University of Texas Rio Grande Valley, Brownsville, TX 78520, USA

**Keywords:** graphene oxide, synthesis, high yield, oxidant reagents, bench reactor

## Abstract

The detailed study of graphene oxide (GO) synthesis by changing the graphite/oxidizing reagents mass ratios (mG/mROxi), provided GO nanosheets production with good yield, structural quality, and process savings. Three initial samples containing different amounts of graphite (3.0 g, 4.5 g, and 6.0 g) were produced using a bench reactor under strictly controlled conditions to guarantee the process reproducibility. The produced samples were analyzed by Raman spectroscopy, atomic force microscopy (AFM), x-ray diffraction (XDR), X-ray photoelectron spectroscopy (XPS), Fourier-transform infrared spectroscopy (FTIR) and thermogravimetry (TGA) techniques. The results showed that the major GO product comprised of nanosheets containing between 1–5 layers, with lateral size up to 1.8 µm. Therefore, it was possible to produce different batches of graphene oxide with desirable physicochemical characteristics, keeping the amount of oxidizing reagent unchanged. The use of different proportions (mG/mROxi) is an important strategy that provides to produce GO nanostructures with high structural quality and scale-up, which can be well adapted in medium-sized bench reactor.

## 1. Introduction

Until graphite monocrystals with atomic thickness were isolated in 2004 by Geim and Novoselov [1], it was accepted that such structural isolation was impossible due to the thermodynamical instability of these nanoparticles. Nowadays, not only their structural isolation at environmental temperature is possible, but also some of their characteristics are known, such as high charge mobility (~10,000 cm^2^ V^−1^ s^−1^) [1,2] and electrical conductivity (7200 S m^−1^) [3]. In this context, graphene is considered a nanomaterial formed by a honeycomb-like structure with thickness of one carbon atom, very important for many applications in a several of fields [4,5,6,7,8]. Despite its large field of application, the large-scale production of graphene is still challenging, due to the high cost of the process, therefore, as an alternative, graphene derivatives have been developed [9,10,11,12]. Although studies on graphite oxidation have been carried out for a different purpose, methods for that are known in the literature since the 19th century. With the acknowledgement of graphene oxide (GO) happening in the 21st century, such methods have been adapted to its production. In the last decade, the method developed by Hummers in 1958 [13] has been modified and widely used by several researchers to obtain graphene oxide [14,15], thus several modifications were made to improve this technique and its yield [16]. Studies on the use of different amounts of oxidizing agents [17,18], degree of oxidation [19], influence of reagents on the oxidation process [17] and structural quality of the nanosheets, were presented [9,15,16,20,21,22,23]. Due to the different methods of obtaining graphene and its derivatives, in 2017 an ISO/TS 80004-13 (International Organization for Standardization, 2017) was created, listing the terms and definitions for graphene and the other two-dimensional carbon materials. Even with many studies exploring variations on the GO production, reports lack information about the parameters and employed equipment [15]. In literature, it is discussed that the amounts of reagents are already close to the saturation limit, but it is known that factors such as temperature or stirring are directly related to the chemical reaction kinetics, and the precise control of these factors interfere in the reaction yield. Although these processes are already in use, aspects such as how to minimize produced waste and reagents used, reproducible methods and quality control are factors that still need to be explored in the large-scale synthesis of GO. A good strategy to improve the GO production is to reduce the amounts of oxidizing reagents used while maintaining the same experimental conditions of the process.

In this context, the aim of this work is to obtain GO nanosheets from a bench reactor with minimal amounts of reagents, using the best possible mass ratio between graphite and oxidizing reagents (mG/mROxi), by using the improved Hummers methods [14]. The application of a set of techniques (XRD, TGA, FTIR, Raman spectroscopy and AFM) was suitable to highlight the choice of the best condition to produce a material with graphene oxide structural features and high yield.

## 2. Materials and Methods

### 2.1. Materials

Synthetic graphite powder <20 µm (C(s)), sulfuric acid 95–98% (H_2_SO_4_), potassium permanganate (KMnO_4_), hydrochloric acid 37% (HCl) used herein were supplied by Sigma-Aldrich Brazil-Ltda. Hydrogen peroxide (H_2_O_2_) 35% was supplied by Labsynth Ltd.a. (Diadema, Brazil) and phosphoric acid (H_3_PO_4_) 85% was supplied by Vetec Brazil Ltd.a (Rio de Janeiro, Brazil).

### 2.2. Methods and Experimental

The improved method to produce GO was based on the work of Marcano et al. [14]. All samples were synthetized using a bench reactor (Mettler Toledo model RC1e). Due to the high equipment precision in relation to the reaction parameters control (such as trend of heat flow and rotation speed), it is possible to obtain favorable conditions that increase the efficiency of graphite oxidation, without major thermal variations of the system. In this way, the reproducibility of the GO production in the reaction is guaranteed, as all these stages are strictly controlled. To evaluate the influence of graphite/oxidizing reagent mass ratios (mG/mROxi), three different samples were prepared where the quantity of the oxidizing reagents (such as H_2_SO_4_/H_3_PO_4_ and KMnO_4_) was maintained, varying only the mass of graphite (3.0 g, 4.5 g, and 6.0 g). In the first step, 360 mL of H_2_SO_4_ was added to the reaction vessel, keeping the jacket temperature at 5 °C and low rotation. Under constant stirring, 40 mL of H_3_PO_4_ was added maintaining the mentioned temperature. Next, different amounts of graphite were added for each sample as mentioned in Appendix A. The reaction medium was homogenized, and the temperature was reduced to 0 °C. Then, 18 g of KMnO_4_ was added, slowly and gradually to avoid excessive heating, since this stage is extremely exothermic. The reaction medium was homogenized for 1 h, keeping temperature and stirring constant. The reaction system was heated to 50 °C, temperature and stirring where kept constant for 12 h. After this period, the reaction medium was cooled down again to 0 °C with the addition of 1.0 L of cold deionized water. Hydrogen peroxide (H_2_O_2_) was added to the reaction medium with a pipette. The volume used was based on the color change (Appendix A). As soon as the color changed, as it generally occurs in “titration processes”, the addition of H_2_O_2_ was interrupted and the synthesis in the reactor was finished. The reaction yield for all produced samples was ~130%, considering the increase in mass as a function of the oxidative process. Further details on the production of the different samples of GO nanosheets are also described in (Appendix A), where an additional batch was produced from 9.0 g of graphite with GO multilayer characteristics was also produced.

#### GO Characterization

SEM micrographs were obtained on a Quanta 200 equipment, model FEG FEI 2006 (Sao Paulo, SP, Brazil), operating under vacuum with the electron beam under acceleration voltage between 5 and 30 kV. The GO nanosheets were fixed to a sample holder with the aid of a carbon conductive tape. X ray diffractograms were obtained in a Rigaku Miniflex II (Sao Paulo, SP, Brazil) operating with copper filament tube (λ = 1.42 Å), acceleration speed of 4 kV, scan range from at 3° to 40° with steps of 0.020° and a scan rate of 2° min^−1^. 

Thermogravimetric Analysis (TGA), (Sao Paulo, SP, Brazil) was performed on a SDT Q600 analyzer by TA Instruments, samples were deposited on an alumina crucible, under a synthetic air atmosphere, A heating ramp of 5 °C min^−1^ with a temperature sweep from 35 to 1000 °C was used.

Fourier transform Infrared Spectroscopy (FTIR) techniques were performed on dried GO samples to confirm oxidation, the spectra were obtained in FTIR equipment model IRAffinity-1 Shimadzu, (Sao Paulo, SP, Brazil) operating with ATR module (attenuated total reflectance). The equipment software was configured to carry out 64 transmittance scanners, with a resolution of 4 cm^−1^ and with a scan range from 600 cm^−1^ to 4000 cm^−1^. 

For Raman spectroscopy, a WiTec Alpha 300R Confocal Raman Microscope (Sao Paulo, SP, Brazil) was used equipped with a laser source with λ = 532 nm, the samples were prepared from the 1 mg mL^−1^ GO dispersion, depositing a drop on a silicon oxide substrate and drying it. The process was repeated several times until the substrate was visibly covered with a film of GO. XPS spectra were obtained using monochromatic Al Kα radiation (1486.6 eV) and an electron energy analyzer (Specs, Phoibos-150), (Belo Horizonte, MG, Brazil) that enables high energy resolution and excellent signal to noise ratio. The signal of adventitious carbon (C 1s at 284.6 eV) was used to correct the binding energy scale of the survey and the high-resolution spectra. The detailed oxidation states of the elements were obtained from fitting the XPS peaks assuming their shape as a convolution of Lorentzian and Gaussian functions.

AFM was performed on a Bruker Dimension Icon, with a ScanAsyst-Air probe (Sao Paulo, SP, Brazil). The sample preparation for the AFM images consisted of a dried drop of GO dispersions (previously exfoliated mechanically and kept for 5 days in the desiccator for drying) diluted 200 times onto a fresh mica substrate. It was used equipment from the Bruker manufacturer, model Dimension Icon with ScanAsyst-Air probe. Evaluation of generated images was performed using Gwyddion software.

## 3. Results and Discussion

Figure 1a–c shows SEM images of the produced graphene oxide. The samples have similar morphology with wrinkled and overlapping sheets and lateral sizes up to 10 µm. As it will be observed later through different analytical techniques, nanosheets of GO 3.0, GO 4.5 and 6.0 g have shown to have very similar physical and chemical properties.

By XRD analysis, it is possible to verify the change in the interplanar distance between the graphite layers compared to the starting material. As shown in the diffractogram in Figure 2a, graphite shows a diffraction peak 2θ = 26.4° for the crystallographic planes (002), with an interplanar distance of 0.319 nm [24]. The results show that there was an increase in the distance between the graphite planes for all the GO produced. After the oxidative process, the values of 2θ were lower for all samples. The peak in (002) disappears in GO and a new peak (001) at a smaller 2θ value (greater spacing between sheets) appears, the interplanar distances with their respective displacements can be seen in Table 1. It is known that, after the oxidation process, oxygenated groups are attached on graphite basal planes increasing its interplanar distance, which shifts the reflection peak 2θ to smaller values as a consequent reduction of van der Waals interactions that keep these layers stacked [25]. Through these results, it can be considered that the variation in (mG/mROxi) did not prevent the insertion of oxygenated groups between basal planes, however a comparison with the other analyzes is needed to determine whether this consideration is true. Figure 2b shows the XPS survey scan of the graphite and GO 4.5 sample. XPS data were identified using the binding energy of C 1s at 284.5 eV and O 1s at 531.0 eV, this result will be discussed in detail soon.

Detailed FTIR spectra of GO nanostructures are discussed this section (Figure 3a–d). FTIR spectrum indicates the presence of different functional groups that are present in a sample. Based on the graphite oxidation route it is possible to estimate the oxygenated groups that may be present in the graphitic structure. The first strong and broad bands (1, 2 and 3) are usually attributed to the stretching mode of O-H groups specially derived from carboxylic acids and intramolecular alcohols [26]. The presence of these alcohols is further evidenced by bands 8 and 9 that indicate the presence of stretching modes of C-O in alcohol groups, being 8 for tertiary alcohols within the GO layer, and 9 for the secondary ones that are more likely to appear on the layer edges [26]. Bands 4 and 5 demonstrate the stretching mode of C-H in aldehydes, while 5 the C=O stretching of esters and other carbonylic groups. The band 6 shows the natural characteristic of GO to contain C=C stretching mode of alkenes from the hexagonal network [26,27]. The epoxide groups are represented by the appearance of bands 7 and 10 [14,26]. Given the FTIR spectra are normalized, the larger bands from oxygenated groups in GO 3.0 sample indicates a slightly higher degree of oxidation and/or concentration of oxygenated groups and therefore layer defects. The FTIR spectra obtained show that all samples were oxidized regardless of the reaction conditions. Such modification is important because, it separates the graphitic layers and makes the subsequent dispersion of graphene oxide in water and other polar solvents easier due to increased polarity [26]. Further, it allows the functionalization of GO with compatibilizers, enabling its dispersion in different mediums, such as organic solvents and polymeric matrixes [6]. The main FTIR bands and its respective wavelengths of GO samples are described in the Appendix A.

Likewise, the XPS spectra for all samples also did not show significant variations, detailed high resolution photoemission peaks for C and O atoms of graphite and GO 4.5 samples are shown in Figure 4. The photoemission C 1s peaks were studied between 280–295 eV. Figure 4a shows a peak at approximately 284.6 eV regarding the sp^2^ hybridized C-C bonds in extensive π-π* conjugated systems [28]. A secondary peak is verified at 285.1 eV, which is characteristic of sp^3^ hybridized C-C bonds present at defective locations and structure asymmetry. Photoemission peaks also observed at 285.6, 286.3, 286.8, 286.6 and 287.1 eV are attributed to different carbon atoms bonded to oxygen atoms (-C-O-R), and the others are characteristic of carbon atoms pertaining to carbonyl groups (-C=O) [29]. In ~291.0 eV, a satellite peak is observed, which is caused by the π-π* electronic transition [30]. The evidence of carboxylic groups presence is obtained by the observation of a peak at 288.8 eV (Figure 4c), which is typical for this functional group. The adjustments of the photoemission peak for O 1s are shown in Figure 4b–d. Peak at 532.1 eV is assigned to the C=O bonds, at 533.9 eV can be assigned to the C-OH bond of water, adsorbed on the surface of the material. After the oxidation process, peaks related with binding energies of 532.7, 534.6 and 536.5 eV represent the C=O, C-O-C and C-OH bonds respectively [26].

Figure 5 shows the Raman spectra for the graphite and GO samples. As showed in Figure 5a, the graphite presents a high G band intensity (1574 cm^−1^) and low intensity D band (1348 cm^−1^), characterizing a graphitic structure with few defects. A significant increase in the D band intensity is observed in the GO samples produced when compared to graphene, which indicates that there was a break in the hybridized sp^2^ carbon bonds, a result of the generation of defects and the increase in the number of edges from the breakage of the graphite sheets [31]. As seen in Figure 5b–d, the G bands were smoothly displaced to 1591, 1589 and 1594 cm^−1^, respectively. The D band increase indicates the occurrence of oxidation, since intact crystalline structures such as graphite and graphene generally do not have a sharp or intense D band and the ratio between the relative intensities of these bands (ID/ IG) is smaller than in oxidized graphitic materials [24]. The samples produced showed ID/IG ratios close to 1.0, as seen in Table 2, indicating the existence of defects in their structure, thus proving the formation of GO nanosheets [21,22].

Through the thermogravimetric (TG) and its differential (DTG) curves, (Figure 6a,b), it was possible observe the main thermal events related to the loss of mass of the samples. Regarding the graphite used as a starting material, it was observed that this material remained stable until it presented a mass loss at 822 °C. This high thermal stability demonstrates high crystallinity and low degree of impurities [26]. For the GO samples, three main regions of mass loss were observed. A first one, at 25–100 °C, is related to water molecules or some other adsorbed solvent mass loss. Then, at 100–300 °C, another process refers to the exit of oxygenated groups covalently bonded with the graphitic material, with a maximum thermal degradation at ~199 °C. The last mass loss process occurred at 400–700 °C, due to the thermal decomposition of graphitized structure, with a maximum peak observed at ~592 °C [26,28,32]. The minimal residual mass can be attributed to possible impurities generated by the formation of metallic oxides from the salt used in the in the oxidation process [28]. In the same way that it was observed in the FTIR, XPS and Raman spectra, no significant changes were observed in the thermal behavior of produced samples, showing that the oxidative process under the conditions used was homogeneous for all of them. The respective loss masses percentages in relation to the increase in temperature are summarized in Table 3.

AFM images allow the visualization of the material surface through the acquisition of topographic and phase contrast images. The height of GO nanosheets can be directly linked to the number of layers they have. From the collected AFM images, it was noted that there were differences in the lateral size and thickness. Figure 7a shows the GO 3.0 sample profile, where micrometric sheets were observed with a well-defined shape at the edges, with few folds and layer overlays. GO 3.0 sample showed layers with thicknesses up to 1.2 nm characteristic of bi or three layers [31], with lateral sizes predominantly between 0.5–1.2 µm. AFM image of GO 4.5 sample (Figure 7b), showed nanosheets with thicknesses up to 0.9 nm, containing 2 layers approximately, where sheets of up to ~1.8 µm in lateral size were observed. The same behavior was verified for the GO 6.0 sample, where there is a large amount of bilayer and tri-layered GO, with lateral size up to than 1.0 µm. In all cases, the presence of a big amount of bi-layers was observed. A change in the conformation of the image of this sample was detected due to a small distortion in the AFM tip. However, it can be verified by SEM image (Figure 1c), that the GO 6.0 g nanosheets have the same morphology as the others.

The association of Raman spectroscopy and AFM results, it was clear that the same proportion of agents added, was efficient to produce all the samples, despite of the amount of graphite to be oxidized (3.0, 4.5 or 6.0 g). The relationship between the D and G bands (ID/IG ratio) showed by Raman spectra of these samples have practically the same number of defects, while the AFM images showed similarity between their sizes and morphology. Other similarities among the samples could be as well noticed in the other analyses mentioned.

## 4. Conclusions

This work demonstrated that changing the mG/mROxi ratios with increase of graphite mass up to 6.0 g was viable to produce graphene oxide nanosheets with few layers. This strategy proved to be an alternative for performing GO production with a reduction of oxidizing agents by using a bench reactor under controlled conditions. XRD showed the increase in the distance between the graphite layers of graphite after the oxidation process. In addition, analyzing the SEM and AFM images, it can be said that there was a formation of GO with few layers in all three samples, (1–4 layers), with structural quality. TGA results showed mass loss in the regions belonging to the decomposition of oxygenated groups, whose presence was confirmed by FTIR and XPS. Thus, it was observed that the controlled oxidation process was succeeded with minimal variations in the composition, structure, and morphology for these three different batches. Further investigation varying the mG/mROxi ratios above the values studied and under the same conditions should be carried out to define a limit ratio that can keep reaction efficiency and quality for the GO nanosheets produced. This issue still needs to be explored further.

## Figures and Tables

**Figure 1 nanomaterials-11-01975-f001:**
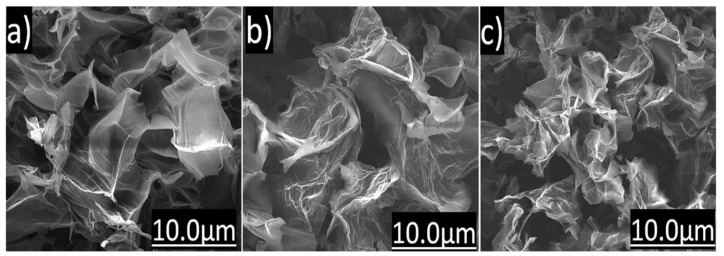
SEM image of (**a**) GO 3.0 g, (**b**) GO 4.5 g and (**c**) GO 6.0 g samples.

**Figure 2 nanomaterials-11-01975-f002:**
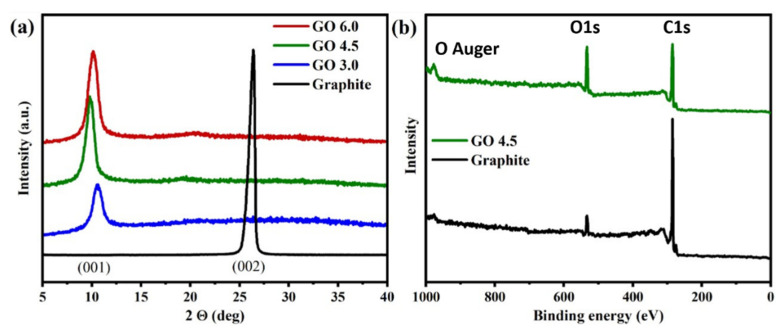
(**a**) XRD profile for GO 3.0 g, GO 4.5 g and GO 6.0 g samples. (**b**) XPS survey spectra of graphite and GO 4.5 sample.

**Figure 3 nanomaterials-11-01975-f003:**
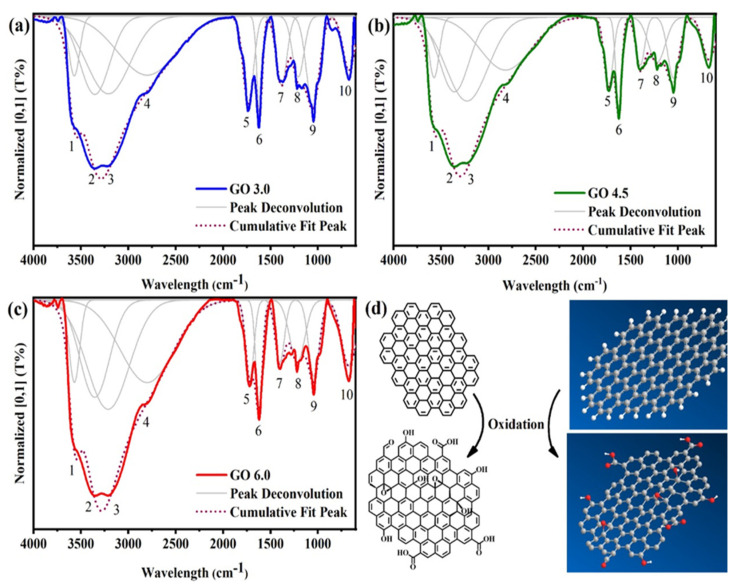
FTIR spectra for samples of (**a**) GO 3.0 g, (**b**) GO 4.5 g, (**c**) GO 6.0 g and (**d**) a representation of the structural modification on the graphitic layer induced by oxidation.

**Figure 4 nanomaterials-11-01975-f004:**
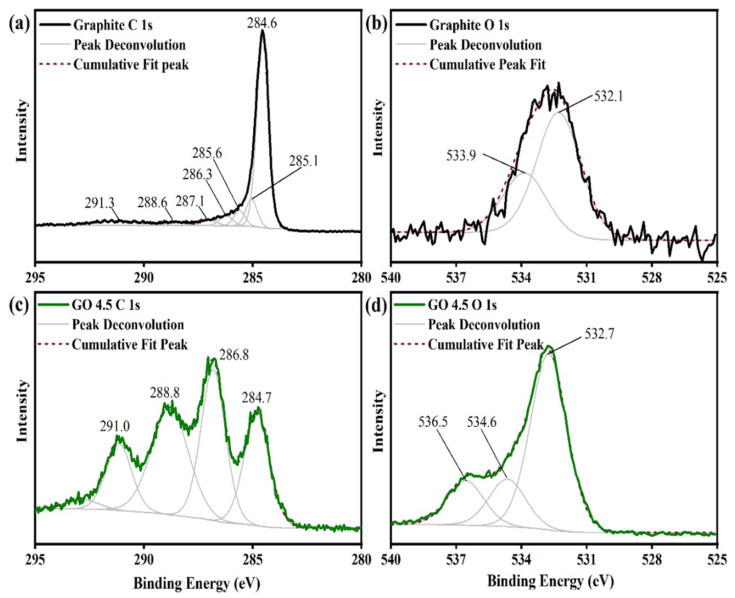
XPS spectra with high resolution photoemission peaks for the C 1 s and O 1 s of graphite (**a**,**b**), and GO 4.5 g sample (**c**,**d**).

**Figure 5 nanomaterials-11-01975-f005:**
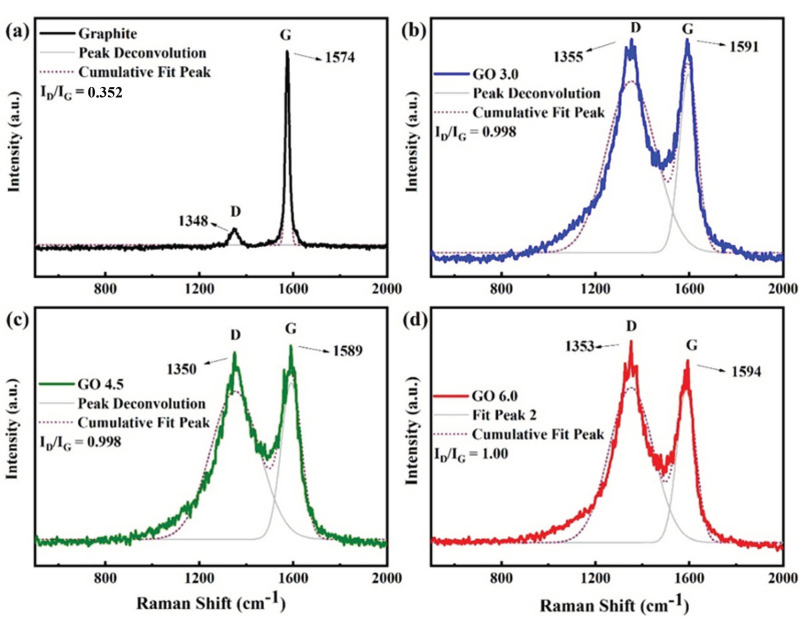
Raman Spectra for samples of (**a**) graphite, (**b**) GO 3.0 g, (**c**) GO 4.5 g and (**d**) GO 6.0 g.

**Figure 6 nanomaterials-11-01975-f006:**
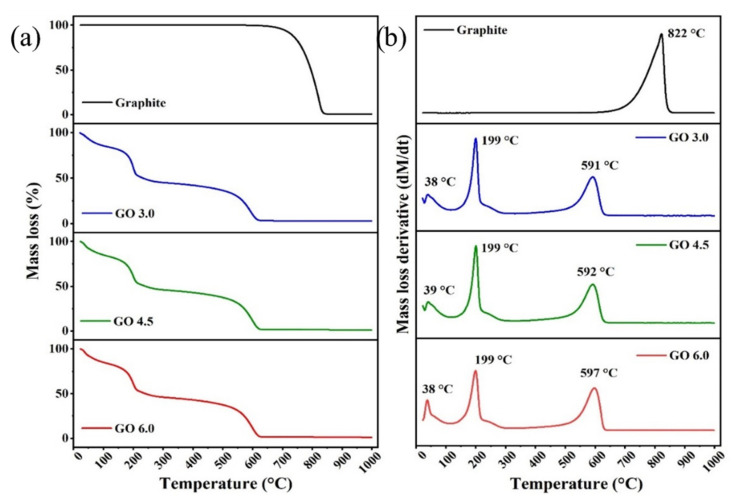
(**a**) TGA and (**b**) DTG curves of graphite, GrO 3.0 g, GrO 4.5 g and GrO 6.0 g samples.

**Figure 7 nanomaterials-11-01975-f007:**
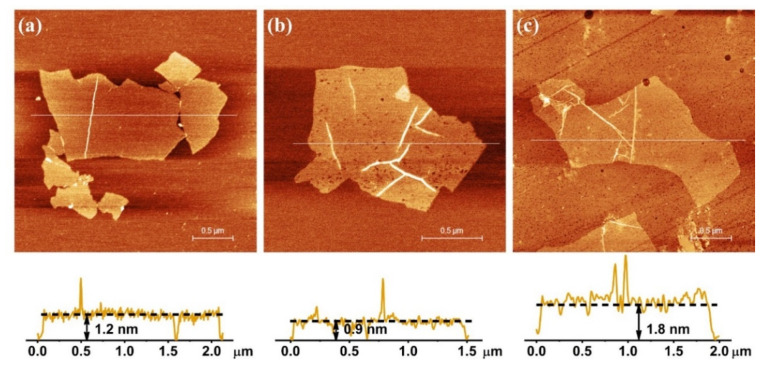
AFM images of (**a**) GO 3.0 g, (**b**) GO 4.5 g and (**c**) GO 6.0 g samples, and its respectively height profiles.

**Table 1 nanomaterials-11-01975-t001:** 2θ values of graphite and GO produced, and its interplanar distances.

Sample	2θ	Δ2θ	Distance (nm)
Graphite	26.4°	0.0	0.319
GO 3.0 g	10.6°	15.8	0.775
GO 4.5 g	9.8°	16.6	0.837
GO 6.0 g	10.1°	16.3	0.813

**Table 2 nanomaterials-11-01975-t002:** Wavenumber (λ), D and G bands intensities and ID/IG ratios of GO samples.

Sample	D Band λ (cm^−1^)	D Band Intensity (a.u.)	G Band λ (cm^−1^)	G Band Intensity (a.u.)	I_D_/I_G_
GO 3.0 g	1355	918	1591	920	0.99
GO 4.5 g	1350	890	1589	891	1.00
GO 6.0 g	1353	911	1594	909	1.00

**Table 3 nanomaterials-11-01975-t003:** Mass loss percentages in relation to different temperature ranges.

Sample	Mass Loss % 25–100 °C	Mass Loss % 100–300 °C	Mass Loss % 400–700 °C	Mass Loss % above 700 °C	Residual Mass Loss %
Graphite	0	0	0	100.00	-
GO 3.0 g	15.00	40.50	42.00	-	2.90
GO 4.5 g	15.00	39.00	44.40	-	1.60
GO 6.0 g	15.00	39.00	44.40	-	1.60

## Data Availability

The data is available on reasonable request from the corresponding author.

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
