# Peer review of "Micro Scalable Graphene Oxide Productions Using Controlled Parameters in Bench Reactor"

_nanomaterials, 2021, doi:10.3390/nano11081975_

Round 1

Reviewer 1 Report

This manuscript shows the synthesis and characterisation of 3 different batches of graphene oxide (GO) produced using different amounts of graphite precursor and a fixed amount of potassium permanganate as the oxidising agent.

The authors have used the well known ‘Tour Method’ as their procedure for synthesising the batches of Go using highly controlled conditions within a bench reactor. A large array of techniques have been used in the study: SEM and AFM, IR and Raman, XPS and XRD, and thermogravimetric analysis.

The figures are very nicely presented and the results have been analysed very thoroughly and properly. The figures are also annotated in a way that helps the reader greatly with interpretation, and complementary tables are provided throughout.

I have only a few queries and suggestions that I would like the authors to address before publishing this article. Firstly, the authors state in the abstract that it is possible to control the physicochemical properties of GO using their approach. However, minimal difference was repeatedly noted throughout the paper in such properties. I believe this is because the amount of proportional permanganate used in the original Marcano paper is by design, vastly in excess to ensure full oxidation of the GO. I therefore believe that the authors should also perform their experiments at least one more much higher quantity of graphite (perhaps 12 g) to show when the oxidation levels are significantly less than the current batches they’ve produced (i.e. less than 130% yield). This would be very useful information for the GO community.

Furthermore, in Figure S1, the 3 samples of GO are stated as being the same concentration (1 mg/mL), however appear to be very visually different (darker for the batches produced from higher graphite quantities). If they are indeed the same concentration then this change in optical properties is most likely linked to the degree of oxidation for each sample (reduced GO appears black). I suggest the authors perform UV-vis measurements on a sample of equal concentration from each batch and compare the absorption maxima and absorbances. If the level of oxidation is different then you will notice a shift in the absortion maxima (fully oxidised GO is at 230 nm, reduced GO is at 260 nm).

If there is still minimal differences between batches then this is something that needs to be explained in detail in the Discussion section.

Here are some other more minor things that were noted:

~ What do the authors mean by ‘good structural quality’?

~ Was the GO purified and if so how? This is not noted in the Methods section. There is a huge amount of acid and ions in the synthesis so purification is a vital step in GO preparation.

~Second last sentence of the AFM paragraph in the Methods section: ‘The equipment used was.’ Is this a typo or an incomplete sentence?

~ The AFM figure (Fig. 7) needs to be improved. The contrast is poor in panels ‘a’ and ‘b’ and height profiles should be shown below each image rather than the colour scale as it is too difficult to tell accurately what the thickness of the sheets are in this manner. These can be easily fixed on Gwyddion.

Lastly, some grammatical errors were noted throughout the paper so the authors should give a final thorough read-through before final submission. Aside from these concerns, a very nice paper with beautiful figures. I would be happy to accept for publication after the above has been addressed.

Author Response

"Please see the attachment" - Cover letter and response to reviewers comments.

Reviewer 2 Report

This manuscript is a report on the synthesis and characterization of graphene oxide (GO) through an improved Hummers method. The scope of the study is to develop a protocol that allows for the minimum consumption of reagents.

I like the motivation for this study, and the results could be of interest for the community. However, in its current form the manuscript cannot be published. The study is incomplete. At the present, the authors have simply shown that by selecting three different graphite to oxidizing reagents ratios, the resulting GO exhibit similar features (size, thickness, amount and type of functional groups and so on). However, they have not yet found the minimum quantity of reagents that is necessary to form "good quality" GO from graphite, which was indeed the scope of the study.

Keeping the format of the experiments as they have already done, they should increase the amount of graphite powder (8g, 10g....100g or more) until they find what the maximum amount of graphite powder (or, in other words, the minimum amount of reagents) that produces GO with a certain kind of properties is. 

Author Response

(The authors gave the same response as above.)

Round 2

Reviewer 1 Report

The authors have addressed my comments thoroughly, and where unable to do so, have provided a legitimate explanation. I hope the situation in Brazil related to covid improves as soon as possible.

I am happy to accept the paper in its current form.

Reviewer 2 Report

I do not understand the authors' reply.

I asked what the maximum amount of graphite that can be turned into graphene oxide with the amount of reagents that they use is. To this aim, I asked them to perform additional experiments with increasing amount of graphitic powder.

Their reply is that they have done already some experiments with 9 g of graphite, but the results were not good. What does "not good" mean? If it is to say that graphite did not oxidize, then this is the answer I asked. The authors need to add these "negative" results to make their study more robust and useful to the community. Negative results are not bad results; on the other hand, they can be very valuable to the community.

Round 3

Reviewer 2 Report

I am very sorry to hear about the current health crisis in Brazil.

Author Response

Authors want to thank for reviewer's comments and comprehension. It is definitively something out of our hands. Close researchers have been infected and the less we want is to be exposed to that. We have been working from home with what we have. We hope to get vaccination soon so we can go back to work and to get access to University and Labs.

We have reviewed the english from our manuscript and it will be incorporated in the last submitted version of our work.

Regards,

Thanks again.